# LIN-35 is necessary in both the soma and germline for preserving fertility in *Caenorhabditis elegans* under moderate temperature stress

**Brian P. Mikeworth, Frances V. Compere⬤¤, Lisa N. Petrella⬤***

Department of Biological Sciences, Marquette University, Milwaukee, Wisconsin, United States of America

¤ Current address: Department of Biology, Syracuse University, Syracuse, New York, United States of America

* lisa.petrella@marquette.edu

**Data Availability Statement:** All image and raw scoring files are available from the Epublications database at Marquette University https:// epublications.marquette.edu/mikeworth_2023/.

## Abstract

Maintenance of germline function under stress conditions is crucial for species survival. The germ line in many species is especially sensitive to elevated temperature. We have investigated the role of the pocket protein LIN-35 in preserving fertility in *Caenorhabditis elegans* under moderate temperature stress. We show that *lin-35* mutants display several temperature sensitive germline defects, and more severe reductions in brood size at elevated temperatures compared to wild type. This loss of fertility under temperature stress is primarily due to loss of zygotic, but not maternal, LIN-35. Additionally, we have found that expression of LIN-35 is necessary in both the germ line and soma for the preserving fertility under moderate temperature stress. Specifically, while LIN-35 function in the germ line is required for maintaining fertility in hermaphrodites, broad somatic expression of LIN-35 is also necessary for oocyte formation and/or function under moderate temperature stress. Together, our data add to the emerging understanding of the critical role that LIN-35 plays in preserving tissues against stress.

## Introduction

Pocket proteins generally function in transcriptional repression, which in mammals has been linked to cell cycle regulation and cell fate decisions [1, 2]. In *C. elegans*, the sole pocket protein is LIN-35, which is homologous to the three mammalian family members Rb (retinoblastoma), p107, and p130 [3]. Although LIN-35 is expressed broadly across most tissue types, it has distinct roles in different tissues including the repression of germline gene expression and RNAi pathways in the soma, suppression of nuclear divisions in the intestine, and regulation of apoptosis in the germ line [4–7]. *C. elegans* carrying mutations in *lin-35* also demonstrate a range of synthetic phenotypes (those that appear only when a second gene is also disrupted), which have revealed roles for LIN-35 in regulating vulval cell fate specification, somatic apoptosis, and somatic gonad development [8–10]. Despite the role of LIN-35 in these varied processes, *lin-35* mutants remain viable and fertile at normal culture temperatures. However, they do

**Funding:** L.N.P. R00GM98436 National Institute of Health NIGMS https://www.nigms.nih.gov/. L.N.P R15GM122005 National Institute of Health NIGMS https://www.nigms.nih.gov/ The funders had no role in study design, data collection and analysis, decision to publish, or preparation of the manuscript.

**Competing interests:** The authors have declared that no competing interests exist.

display slowed growth rates and brood sizes [5, 11]. *C. elegans* mutant for *lin-35* show substantial changes in gene expression in the germ line, suggesting that LIN-35 has an intrinsic function in germ cells [12]. However, *lin-35* mutants also show defects in the somatic gonad and intestine [4, 5, 8, 13], both of which contribute to fertility [14, 15]. Therefore, the tissue where expression of LIN-35 is most important for fertility remains unclear.

Fertility is temperature sensitive across many animal species, including in *C. elegans*. The brood size of *C. elegans* peaks at temperatures around culture temperatures of 18–20˚C [16, 17]. As culture temperatures increase from 20˚C to 26˚C, there is a reduction in brood size, but most hermaphrodites remain fertile [16, 18]. However, above these temperatures (between 26˚C and 27˚C) virtually all hermaphrodites become sterile [18, 19]. These phenotypes have been suggested to occur as a result of a decrease in functional sperm as temperature increases, followed by an additional loss of oocyte function at higher temperatures [17–19]. Interestingly, many *lin-35* mutant phenotypes are also temperature sensitive, including the synthetic multivulva phenotype, larval growth (high temperature larval arrest phenotype), misregulation of developmental chromatin compaction, and ectopic germline gene expression [13, 20, 21]. In addition, the brood size of *lin-35* mutants is decreased when grown at 25˚C compared to at 20˚C [5]. However, since wild-type fertility is also decreased across this temperature range, it is unclear if this represents a temperature sensitive phenotype or if *lin-35* mutants are showing the same trend as wild type.

In this study, we show that *lin-35* mutants display temperature sensitive fertility defects. Using somatically expressed LIN-35(+) transgenes to bypass the high temperature larval arrest phenotype, we uncover temperature sensitive decreases in brood size in *lin-35* mutants that are greater than the reductions in brood size seen in wild type. During moderate temperature stress, *lin-35* mutants also have defects in oocyte formation. Finally, we show that while *lin-35* expression in the germ line is most critical for fertility, *lin-35* expression in somatic tissues also plays an important role in maintaining fertility.

## Materials and methods

### *C. elegans* strains and culture

*C. elegans* were cultured on NGM plates seeded with the *Escherichia coli* strain AMA1004 at 20˚C unless otherwise noted. The following strains were used: N2 (Bristol) was used as wild type, MT10430 *lin-35(n745)* (CGC), LNP0044 *petEx1[let-858p::lin-35::GFP + rol-6]*, LNP0043 *lin-35(n745); petEx1[let-858p::lin-35::GPF + rol-6]*, SS0991 *bnEx56[elt-2p::lin-35::GFP + rol-6]* [13], LNP0016 *lin-35(n745); bnEx56[elt-2p::lin-35::GFP + rol-6]*, LNP0082 *bnEx56(elt-2p::lin-35::GFP + rol-6); petEx3(mir786p::lin-35::GFP, myo-3::mCherry)*, LNP0081 *lin-35(n745);bnEx56(elt-2p::lin-35::GFP + rol-6); petEx3(mir786p::lin-35::GFP, myo-3::mCherry)*, LNP0031 *vrIs56 [pie-1p::lin-35::GFP::FLAG::lin-35 3'UTR + unc-119(+)]* (transgene from [22] obtained from the CGC), LNP0022 *lin-35(n745); vrIs56 [pie-1p::lin-35::GFP::FLAG::lin-35 3'UTR + unc-119(+)]* LNP0038 *vrIs93[mex-5p::lin-35::GFP::FLAG::lin-35 3'UTR + unc-119(+)]* (transgene from [22] obtained from the CGC), LNP0023 *lin-35(n745); vrIs93[mex-5p::lin-35::GFP::FLAG::lin-35 3'UTR + unc-119(+)]*, LNP0042 *+/hT2 [bli-4(e937) let-?(q782) qIs48] (I;III)*, SS0911 *lin-35(n745) /hT2 [bli-4(e937) let-?(q782) qIs48] (I;III)* [13], and LW697 *ccIs4810 [lmn-1p::lmn-1::GFP::lmn-1 3'utr + (pMH86) dpy-20 (+)]* (CGC). Some strains were provided by the CGC, which is funded by NIH Office of Research Infrastructure Programs (P40 OD010440).

### Transgenes

We generated a strain containing pan-somatic expression of LIN-35(+) tagged with GFP from the same plasmid used to create the intestine-specific expression of LIN-35(+) by swapping the

5' promoter sequences (*elt-2p* to *let-858p*) using the 2100 base pairs directly upstream of the *let-858* gene. Wild type (N2) strains were microinjected with the *let-858p::lin-35::GFP* plasmid and pRF4 to create the extrachromosomal array *petEx1[let-858p::lin-35::GFP + rol-6]*. When the transgene was crossed into *lin-35(n745)* mutants, the *lin-35(n745)* allele was confirmed via random fragment length polymorphism (RFLP) and western blot analysis.

## Temperature treatments

Three temperature treatments were used in this study. (1) Continuous exposure to 20°C: experiments were performed on strains maintained continuously at 20°C. (2) Continuous exposure to 26°C: P0 hermaphrodites were up-shifted to 26°C at the L4 stage and experiments were done on F1 hermaphrodites that had experienced their entire lifespan at 26°C (3) Up-shift from 20°C to 26°C: hermaphrodites were developed at 20°C until the L4 larval stage and then up-shifted to 26°C and experiments were done on the upshifted P0 worm.

## Western blot analysis

Eighty F1 progeny raised at either 20°C or 26°C were collected for total protein samples in wild type, *lin-35(n745)*, and strains containing somatic *lin-35::GFP* transgenes. Total protein from the entire sample was run on a 7.5% SDS-PAGE gel and transferred to a Nitrocellulose membrane. Membranes were blocked (5% non-fat dry milk in PBS) and incubated in 1:1000 Rabbit anti-LIN-35 [23] in block and 1:200 Mouse anti-TUBB-E7 (Developmental Studies Hybridoma Bank, University of Iowa, Iowa City, IA, USA) in block overnight at 4°C. Membranes were blocked a second time and incubated with 1:2000 Goat anti-rabbit-HRP (Thermo-Fischer Scientific, USA) and 1:2000 Goat anti-mouse-HRP (ThermoFischer Scientific, USA) conjugated antibodies for one hour in the dark at room temperature. Blots were visualized via HRP-chemiluminesence (Amersham RPN2232) and exposure to X-ray film.

## Brood size assay

To assay brood size, 10–15 L4 hermaphrodites were cloned out onto individual plates and placed at the assay temperature based on the temperature treatment protocol being used (see above). Each hermaphrodite assayed was moved to fresh plates daily until no embryos were seen on plates. F2 progeny were counted during the L4 larval stage or as young adults. Brood size assays of mated hermaphrodites were conducted following a similar protocol except that 3 N2 males raised at 20°C were added to the plate. All hermaphrodites and males were transferred to seeded NGM plates daily, with the addition of three new N2 males raised at 20°C on day 3 post-initial mating. Statistical analysis was performed with exclusion of sterile hermaphrodites using either two-way ANOVA or students T-test in GraphPad PRISM (GraphPad Software Inc., La Jolla, CA, USA).

## Maternal effect

Maternal effect of LIN-35 was observed through the use of the hT2 balancer tagged with GFP. P0 *lin-35(n745)/hT2::GFP* where plated at 20°C and upshifted to 26°C at the L4 larval stage. By cloning out *lin-35(n745)/hT2::GFP* or *lin-35(n745)* F1 progeny at the L4 larval stage, we were able to test M+Z+ and M+Z- respectively. To test for M-Z+, *lin-35(n745)* mothers were mated to LW697 *ccIs4810 [lmn-1p::lmn-1::GFP::lmn-1 3'utr + (pMH86) dpy-20(+)]* males at 26°C, where GFP(+) F1 M-Z+ progeny were selected and clone out at the L4 larval stage. Brood size for each cloned out F2 was counted as described above.

## Sperm localization assay

Sperm localization was observed by mating N2 males stained with MitoTracker Red CMXRos (Invitrogen) with mutant hermaphrodites. P0 mothers were plated at 20˚C or upshifted to 26˚C at the L4 larval stage. Males raised at 20˚C were stained with 50μL of 50μM Mitotracker Red CMXRos in M9 for 2–4 hours in the dark. 25 stained males were placed onto NGM plates seeded with HB101 *E. coli* containing approximately 20 F1 L4 hermaphrodite larvae (raised at 20˚C or 26˚C) and allowed to mate for 12–16 hours. Mated hermaphrodites were transferred to a clean plate for 1 hour before imaging to allow any recently deposited male sperm to localize within the germ line. Worms were imaged on a 2% agarose pad in 10mM Levamisole. Normarski and fluorescent images were acquired using a Nikon Eclipse TE2000-S (Nikon Instruments Inc., Elgin, IL, USA) inverted microscope at 60X with Q-Capture Pro (QImaging, Surrey, BC, Canada) imaging software. Sperm were scored as either primarily localizing within the spermatheca, a mixture of localization to the uterus and spermatheca, or primarily localized within the uterus.

## Oocyte counting

Adult hermaphrodites were imaged live using Normarski optics and images were acquired using a Nikon Eclipse TE2000-S (Nikon Instruments Inc., Elgin, IL, USA) inverted microscope at 60X with Q-Capture Pro (QImaging, Surrey, BC, Canada) imaging software. One gonad arm was imaged per hermaphrodite and scored for the presence of one or more than one oocyte in the proximal gonad. Oocytes were defined as cells having a large clear nucleus and taking up the entire width of the proximal gonad.

## Immunohistochemistry

For L1 staining: Young adult P0 mothers were placed in 1XM9 solution without bacteria and allowed to lay embryos overnight at 20˚C. F1 L1 larvae were transferred to slides coated with poly-L-Lysine, freeze cracked, and fixed with methanol/acetone [13]. For adult germ lines: Young adults were transferred without bacteria to slides coated with poly-L-Lysine in 0.1 mM levamisole. Germ lines were dissected before being freeze cracked, and fixed with methanol/ acetone. For both sample types: slides were blocked (1.5% BSA, 1.5% OVA, 0.05% NaN$_3$ in PBS) and incubated with primary antibodies overnight at 4˚C. For L1 animals 1:10,000 rabbit anti-PGL1 [24] was used. For adult germ lines DSHB-GFP-8H11 antibody at 1:100 concentration was used. DSHB-GFP-8H11 was deposited to the Developmental Studies Hybridoma Bank (DSHB) by the DSHB, created by the NICHD of the NIH and maintained at The University of Iowa, Department of Biology, Iowa City, IA 52242. Samples were incubated with secondary antibodies conjugated with Alexa Fluor 568 or 488 at 1:300 (ThermoFischer Scientific, USA) in PBS for 1 hour at room temperature. Slides were treated with 5mg/ml DAPI in 50 mL PBS for 10 min, washed 3 times with PBS for 10 min at room temperature and mounted on gelutol mounting medium. Z-stacks were taken using Nikon A1R Inverted Microscope Eclipse Ti confocal microscope with NIS Elements AR 3.22.09 at 60X.

## Results

### Somatically expressed *lin-35* can rescue high temperature arrest and allow for the analysis of germline temperature sensitivity

Germ line development and function occur primarily after L1 stage, which precludes the direct analysis of temperature influences on the adult germ line in *lin-35* mutants because approximately 100% of these animals display high temperature larval arrest (HTA) at 26˚C [13]. The

native *lin-35* locus is ubiquitously expressed and previous experiments have shown that larval arrest and germline defects can be rescued using the endogenous promoter [22]. To determine if the germline defects observed in *lin-35* mutants are temperature sensitive in a manner similar to somatic *lin-35* phenotypes [13, 20, 21], we used two transgenic strains in which *lin-35* is absent from the germ line, but animals could still bypass the larval arrest phenotype. First, we expressed *lin-35::GFP* from the *elt-2* promoter (*elt-2p::lin-35::GFP)*, which allows for intestinal specific expression of *lin-35(+)* (Fig 1A). We have previously shown that the *elt-2p::lin-35::GFP* transgene in a *lin-35* mutant background can rescue the HTA phenotype, while leaving all non-intestinal tissues mutant for *lin-35* function [13]. We also created a transgenic line that expressed *lin-35::GFP* in all somatic cells under the ubiquitously expressed *let-858* promoter (*let-858p::lin-35::GFP)*. These two *lin-35(+)* transgenes are maintained on extrachromosomal arrays, which are generally silenced in the germ line irrespective of the promoter used [25]. Indeed, our analysis of both of LIN-35(+)::GFP in both L1s and adults revealed no detectible expression in the germ line (S1 Fig) [13]. These two *lin-35(+)* transgenes allowed us to analyze the function of *lin-35* in the germ line during moderate temperature stress.

### *lin-35* mutants display a temperature sensitive decrease in brood size that is only partially rescued by somatic expression of *lin-35::GFP*

To determine if *lin-35* mutants demonstrate temperature sensitive germline defects, we scored both the brood size of individual worms and the percentage of fertile hermaphrodites in populations grown under two different moderate temperature stress treatments when *lin-35* was missing from the germ line. *lin-35* mutants have been previously shown to have a reduced brood size at 20˚C; however, the cause of the reduction is unknown [5]. We confirmed that *lin-35(n745)* mutants grown at 20˚C had a drastically reduced brood size; however, the percentage of fertile worms in the population was only weakly reduced compared to wild type (Fig 1B and 1C). *lin-35* mutants expressing either somatic *lin-35(+)* transgene at 20˚C showed a partial, but significant rescue in brood size compared to the *lin-35(n745)* mutant without a transgene (Fig 1B). These data indicate that the loss of fertility in *lin-35* mutants at 20˚C is partly due to loss of *lin-35* function in somatic tissues.

As temperature increases to 26˚C brood size decreases in wild-type *C. elegans* hermaphrodites, but the percentage of hermaphrodites that are fertile remains close to 100% (Fig 1B and 1C) [18, 19]. To allow for comparison between our transgenic lines and *lin-35(n745)* mutants with no *lin-35(+)* transgene, we grew hermaphrodites at 20˚C until the L4 stage when they were upshifted to 26˚C (upshift). These conditions allowed us to bypass the HTA phenotype seen in *lin-35* mutants because the exposure to 26˚C occurred after most of larval development. The brood size in upshifted *lin-35* mutants with or without a somatically expressed *lin-35(+)* transgene was significantly lower than wild type (Fig 1B). However, unlike our findings at 20˚C, the *lin-35(+)* transgenes could not rescue the brood size compared to *lin-35(n745)* mutants without a *lin-35(+)* transgene (Fig 1B). As the different genotypes displayed different brood sizes at 20˚C, we compared the relative brood size of each strain upon upshift to that same strain's brood size at 20˚C (Fig 1D). We found that *lin-35* mutants with or without somatic *lin-35(+)* transgenes had a stronger relative decrease in brood size compared to the wild type when upshifted to 26˚C (Fig 1D). Similarly, the percentage of fertile worms in the population in upshifted worms was equivalently lower in *lin-35* mutants with or without a somatic transgene compared to wild type (Fig 1C). Overall, the fertility of *lin-35* mutants was more sensitive to increases in temperature at the L4 stage than wild type, and expression of wild-type *lin-35* in the soma did not rescue this temperature sensitivity.

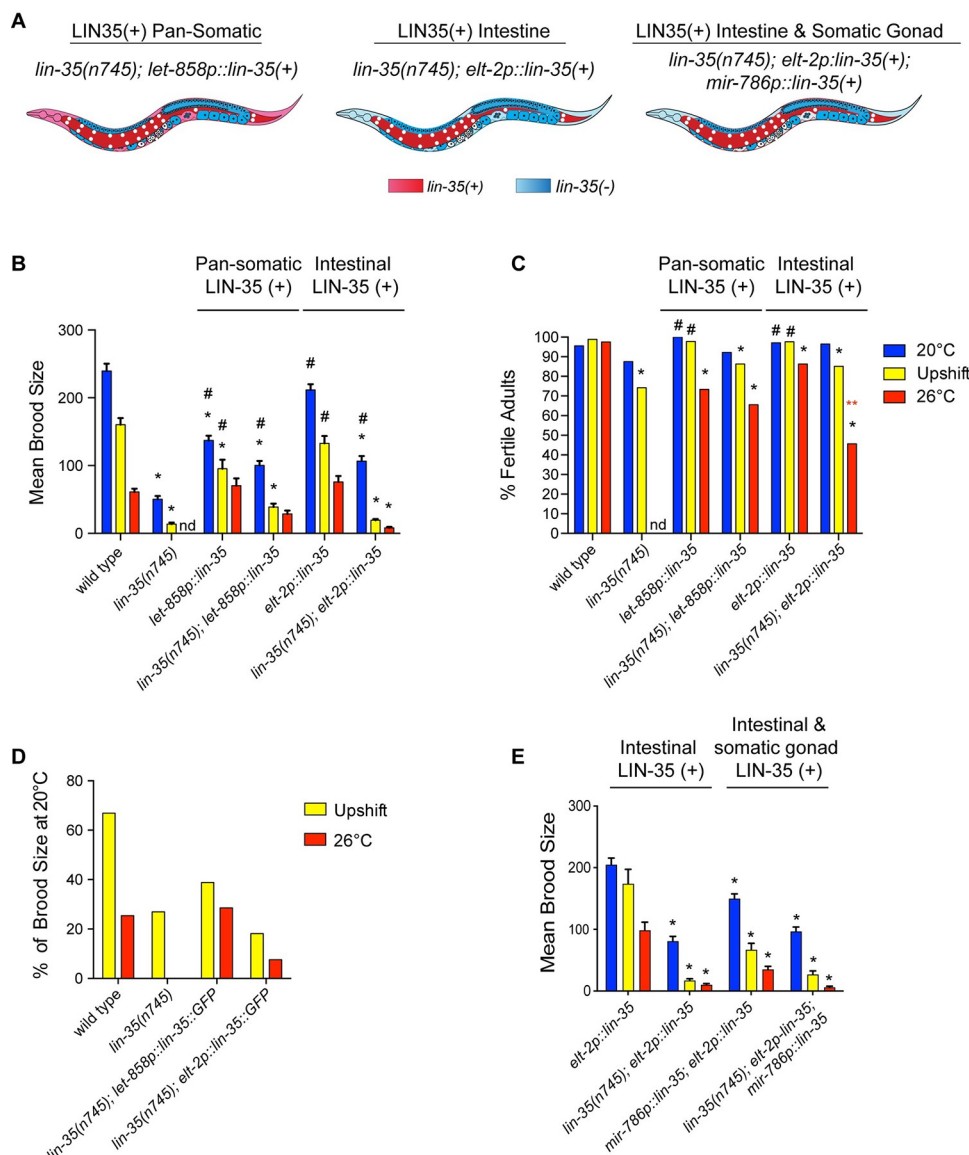

**Fig 1. Somatic expression of *lin-35(+)* partially rescues *lin-35* mutant fertility defects.** (A) Schematic of different somatic transgenes used in this study. Tissues expressing wild type *lin-35(+)* are shown in red/pink, tissues lacking *lin-35* expression *(lin-35(-))* are shown in blue (B) Mean brood size of fertile worms for each genotype across three different growth temperatures (continual growth at 20˚C (blue), upshift from 20˚C to 26˚C at the L4 stage (yellow), and continual growth at 26˚C (red)). Expression of *lin-35(+)* in *lin-35* mutants from either somatic transgene partially rescued brood size at 20˚C, while pan-somatic expression alone rescued brood size at 26˚C. nd: no data, * significantly different than wild type at the same temperature, # significantly different than *lin-35* mutants at the same temperature. *P*-value ≤ 0.05 using two-way ANOVA with Tukey correction. Error bars indicate ± SEM. (C) Percentage of hermaphrodites in the population that are fertile for each treatment. *lin-35* mutants expressing intestinal *lin-35(+)* have significantly fewer fertile worms at 26˚C than those expressing pan-somatic *lin-35(+)*. * significantly different than wild type at the same temperature, # significantly different than *lin-35* mutants at the same temperature, ** (red) significantly different than *lin-35(n745); let-858p::lin-35*. *P*-value ≤ 0.05 using a Fisher's exact test. (D) The decrease in fertility with moderate temperature stress represented by the percentage of the mean brood size remaining with upshift or at 26˚C compared to 20˚C within the same strain. With upshift, *lin-35* mutants show a smaller relative brood size than wild type even if expressing somatic *lin-35(+)*. At 26˚C, *lin-35* mutants with pan-somatic, but not intestinal, expression of *lin-35(+)* retain a similar relative brood size as wild type. (E) Mean brood size of fertile worms for each treatment with expression of *lin-35(+)* in the somatic gonad. *significantly different than *elt-2p::lin-35* at the same temperature. *P*-value ≤ 0.05 using two-way ANOVA with Tukey correction. Error bars indicated ± SEM.

We next investigated the effects on fertility of raising hermaphrodites for their entire development at 26°C. While *lin-35* mutants expressing either somatic *lin-35(+)* transgene showed similar effects on brood size when animals were raised at 20°C, whether upshifted or not, there was a marked difference in brood size between the *lin-35(+)* transgenes in *lin-35(n745)* mutants raised at 26°C. While *lin-35* mutants that expressed the intestinal *lin-35(+)* transgene showed a significantly lower brood size than wild type at 26°C, *lin-35* mutants that expressed the pan-somatic *lin-35(+)* transgene had brood sizes that were indistinguishable from the wild type (Fig 1B). Additionally, the relative decrease in brood size between worms raised at 20°C and 26°C was greater in *lin-35* mutants that expressed the intestinal *lin-35(+)* transgene than those that expressed the pan-somatic *lin-35(+)* transgene (Fig 1D). In fact, at 26°C *lin-35* mutants expressing pan-somatic *lin-35(+)* displayed a similar reduction in brood size to wild type. Finally, the percentage of the population that was fertile was significantly lower in *lin-35* mutants that expressed the intestinal *lin-35(+)* transgene than those that expressed the pan-somatic *lin-35(+)* transgene (Fig 1C). Overall, these data underscore a clear difference in the capacity of pan-somatically expressed *lin-35(+)* to rescue fertility over intestinal-specific expression of *lin-35(+)* during moderate temperature stress.

The rescue of brood size at 26°C in *lin-35* mutants with the pan-somatic *lin-35(+)* transgene, but not the intestinal *lin-35(+)* transgene, suggested that *lin-35(+)* is important in non-intestinal somatic lineages when worms are raised at 26°C. *lin-35* has previously been shown to function in the somatic gonad [8]. Therefore, we expressed *lin-35::GFP* under the *mir-786* promoter, which is expressed in the somatic gonad [26], in combination with intestinal expression of *lin-35(+)*. However, no increase in brood size was observed during moderate temperature stress with expression of *lin-35(+)* in the somatic gonad (Fig 1E). As even pan-somatic expression of *lin-35(+)* could not rescue fertility under all temperature treatments, there is likely a germline intrinsic function for LIN-35 that is necessary for preserving fertility during moderate temperature stress.

In the course of our experiments, we observed that pan-somatic expression of *lin-35(+)* in a wild-type background (i.e. somatic overexpression of *lin-35(+)*) resulted in a reduced brood size compared to wild type without the *lin-35(+)* transgene (Fig 1B). In a *lin-35* mutant background, this *lin-35(+)* transgene fully rescues the HTA phenotype and rescues fertility close to the level of the *lin-35(+)* transgene in the wild-type background at 20°C. One possible explanation for the reduced brood size when *lin-35(+)* is expressed in wild type is that the high levels of LIN-35::GFP driven by the strong *let-858* promoter may display a dominant negative effect in somatic tissue that interacts with the germ line, which is supported by previous work that has shown that LIN-35 can bind its own promoter [22, 23]. Western blot analysis of wild-type animals expressing *let-858p::lin-35(+)::GFP* revealed that the level of native LIN-35 was reduced in the presence of the transgene (S2 Fig). Thus, it is possible that the expression of LIN-35::GFP may down-regulate the native LIN-35 locus.

## Germline expression of *lin-35:GFP* strongly rescues brood size in *lin-35* mutants

To determine if LIN-35 has an intrinsic function in the germ line for preserving fertility at 26°C, we analyzed *lin-35* mutants with germline-specific expression of *lin-35* using two different germline-specific promoters: *pie-1p::lin-35::GFP* and *mex-5p::lin-35::GFP* [22]. Because somatic expression of *lin-35* is necessary to rescue the HTA phenotype, we could only examine the effects of temperature in upshifted L4 hermaphrodites. Expression of *lin-35(+)* from either transgene resulted in partial, but highly significant rescue of brood size at both 20°C and upon upshift from 20°C to 26°C (Fig 2A). Additionally, the relative decrease in brood size between

worms upshifted to 26˚C and worms raised at 20˚C for either germline-specific *lin-35(+)* transgene was equivalent to wild type (Fig 2B). This contrasts with soma-only expression of *lin-35(+)* that resulted in no rescue of brood size and a greater relative decrease in brood size when challenged with temperature upshift (Fig 1D). Therefore, germline intrinsic expression of LIN-35 is primarily responsible for preserving brood size, especially under moderate temperature stress.

## Zygotic *lin-35* expression is sufficient and necessary for fertility levels under moderate temperature stress

We next investigated whether the increased loss of fertility at 26˚C in *lin-35* mutants was a maternal effect phenotype like the HTA [13] and synMuv phenotypes [9]. To achieve this, we tested the fertility of first generation *lin-35(n745)* homozygous M+Z- mutants from heterozygous mothers where *lin-35(n745)* is balanced by the *hT2* translocation that expresses GFP in the pharynx (M+Z-, defined as an animal that has a wild-type *lin-35* maternal load (M+) but is mutant for *lin-35* zygotic expression (Z-)). The *hT2* translocation itself causes a high penetrance of embryonic lethality in a wild-type background [27]. Therefore, we compared *lin-35 (n745)* M+Z- hermaphrodites to wild-type first generation GFP- progeny of +/*hT2* hermaphrodites (Fig 3A). At 20˚C, *lin-35(n745)* M+Z- homozygotes displayed an intermediate brood size between that observed in wild type and *lin-35(n745)* M-Z- homozygous mutant hermaphrodites (M- lacking a maternal load) (Fig 3A). Thus, at 20˚C the decreased brood size in *lin-35* mutants is partially maternal effect. However, at 26˚C *lin-35(n745)* M+Z- hermaphrodites were completely sterile, indicating that loss of just zygotic *lin-35(+)* expression is sufficient to result in a total loss of progeny production at 26˚C.

To further investigate the role of zygotically expressed *lin-35* in fertility we crossed *lin-35 (n745)* M-Z- hermaphrodites with wild-type males that carried a GFP marker and scored the brood size of the resulting *lin-35* heterozygote M-Z+ hermaphrodites. These worms lacked maternally loaded LIN-35(+) protein and mRNA in early embryogenesis, but expressed zygotic LIN-35(+) protein. At 20˚C *lin-35(n745)* M-Z+ hermaphrodites had a significantly smaller average brood size than wild-type hermaphrodites, but a significantly higher average brood size than *lin-35(n745)* M+Z- hermaphrodites (Fig 3A). At 26˚C *lin-35(n745)* M-Z+ hermaphrodites had an average brood size that was not significantly different than wild-type hermaphrodites (Fig 3A). Thus, zygotic expression alone can strongly rescue the fertility of *lin-35* mutants under non-stress temperatures and completely rescue fertility during moderate temperature stress.

## Mating does not significantly rescue fertility in *lin-35* mutants under most conditions

Decreased brood size in wild-type worms at 26˚C is primarily due to decreased sperm function [17–19]. To determine if the significant decrease in brood size in *lin-35* mutants at 26˚C is also primarily due to decreased sperm function, we crossed *lin-35(n745)* hermaphrodites with and without *lin-35(+)* somatic transgenes to wild-type males raised at 20˚C to provide functional sperm. At 20˚C, the number of sperm is the limiting factor in brood size [28]; therefore, crossing with a male to provide more sperm generally leads to an increase in brood size. For hermaphrodites grown at 20˚C, there was a significant increase in brood size in all wild-type strains when mated. However, for strains with a *lin-35(n745)* mutation, only those expressing the pan-somatic *lin-35(+)* transgene showed a significant increase in brood size when mated (Fig 3B). The number of functional sperm made by hermaphrodites is reduced at 26˚C; therefore, crossing with a male raised at 20˚C should also increase brood size if oocyte function is

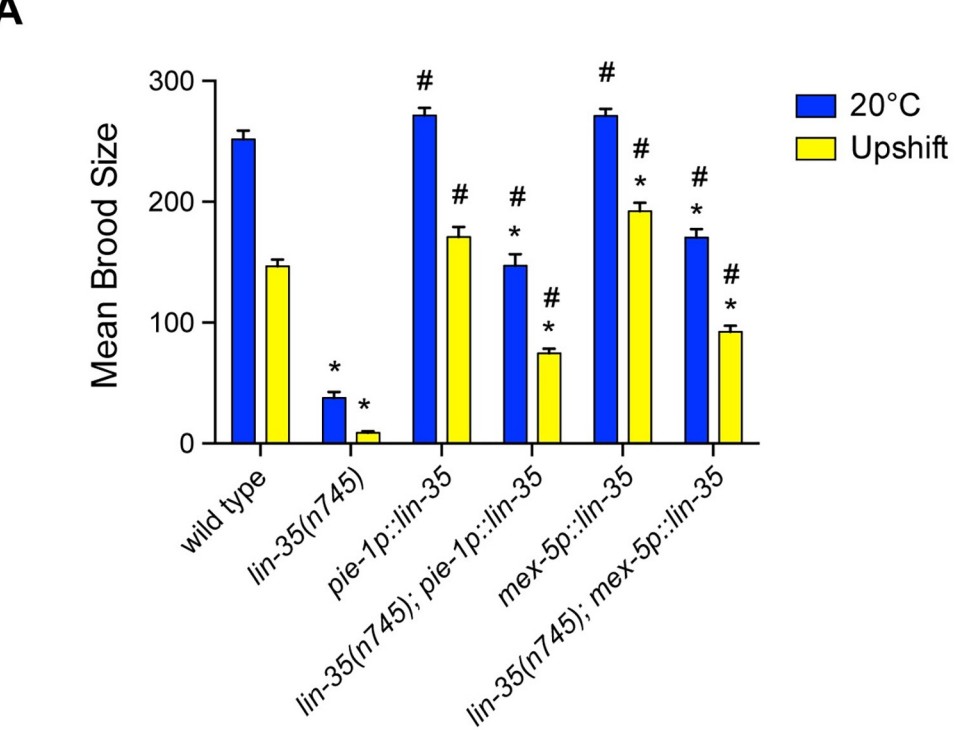

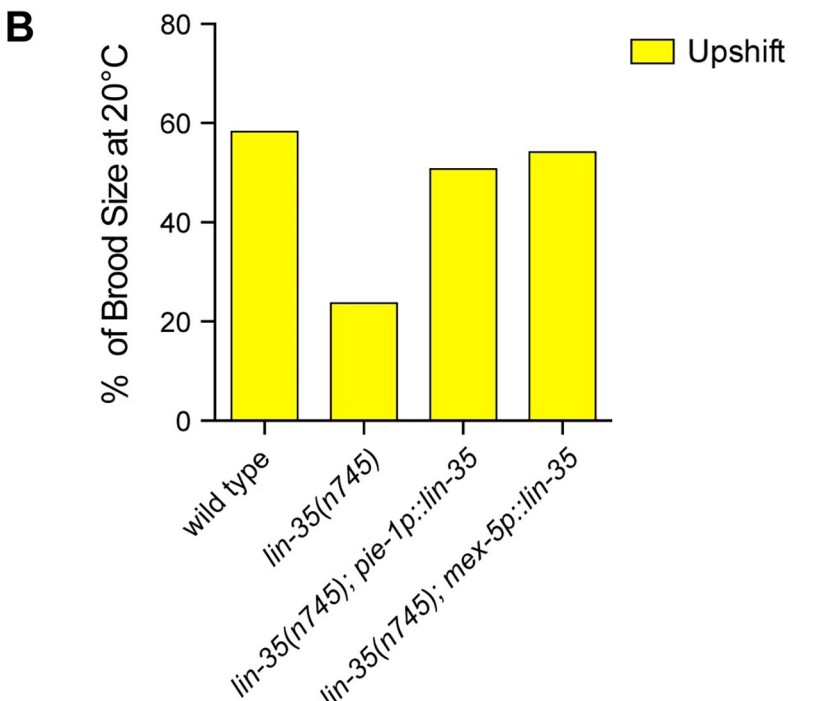

**Fig 2. Germline expression of *lin-35(+)* rescues *lin-35* mutant fertility defects.** (A) Mean brood size of fertile worms for the different genotypes grown under different temperature conditions (continual growth at 20°C (blue) and upshift from 20°C to 26°C at the L4 stage (yellow)). Expression of either germline-expressed *lin-35(+)* transgene significantly

rescued brood size at both 20˚C and with upshift. * significantly different than wild type at the same temperature, #
significantly different than *lin-35* mutants at the same temperature. *P*-value ≤ 0.05 using two-way ANOVA with
Turkey correction. Error bars indicate ± SEM. (B) The decrease in fertility with moderate temperature stress
represented by the percentage of the mean brood size remaining under upshift conditions compared to growth at 20˚C
within the same strain. With upshift, *lin-35* mutants expressing germline *lin-35(+)* retain a similar relative brood size
to wild type.

not compromised. For hermaphrodites grown at 26˚C, no increase in brood size was seen in
*lin-35(n745)* mutants expressing either somatic *lin-35(+)* transgene (Fig 3B). Thus, the reduc-
tions in brood size in *lin-35* mutants are unlikely to be solely caused by dysfunctional sperm.

## Pan-somatic expression of LIN-35(+) is necessary for proper sperm localization under moderate temperature stress

There are two possible scenarios that could lead to the observation of *lin-35* mutants with or
without somatic *lin-35(+)* transgene expression showing limited increases in brood size when
mated. First, when *lin-35* mutant hermaphrodites mate with males there could be limited
sperm transfer/migration to the spermatheca. Second, *lin-35* mutants could have oocyte
defects that limit the ability of male sperm to generate cross progeny [29]. To test the first sce-
nario, we labeled male sperm by incubating males with mitoTracker Red and then mated them
with hermaphrodites for a period of 12–16 hours. We then assessed if sperm were present in
hermaphrodites and able to migrate to the spermatheca (Fig 4A and 4B). Strong localization of
male sperm to the spermatheca was observed in most strains for hermaphrodites raised at
either 20˚C or 26˚C, despite some sperm remaining in the uterus (Fig 4A and 4B). The only
animals without sperm localization to the spermatheca were *lin-35(n745)* mutants with intesti-
nal expression of *lin-35(+)* grown at 26˚C. Overall, these data indicate that sperm can localize
correctly in *lin-35* mutant hermaphrodites raised at 20˚C, suggesting that oocyte defects may
be present in *lin-35* mutants even at 20˚C. However, as male sperm can still localize to the
spermatheca at 26˚C when *lin-35(+)* is expressed broadly in the soma, it is likely that additional
factors contribute to the low brood size of *lin-35* mutants during moderate temperature stress.

## Pan-somatic LIN-35(+) is necessary for oocyte formation under moderate temperature stress

To determine if the loss of fertility in *lin-35* mutants could be due to an inability to produce
oocytes, we quantified the number of oocytes in *lin-35(n745)* hermaphrodites with and with-
out *lin-35(+)* transgenes at 20˚C and 26˚C. Each germ line was analyzed to determine if ani-
mals had more than one oocyte in a single gonad arm (Fig 5A). Close to 100% of gonads had
>1 oocyte in all strains except for *lin-35(n745)* mutants expressing the intestinal *lin-35(+)*
transgene at 26˚C (Fig 5B). Germ lines in *lin-35(n745); elt-2p::lin-35::GFP* hermaphrodites at
26˚C generally looked small, had poor organization, and typically contained only one cell
proximal to the spermatheca that resembled an oocyte (Fig 5A). These data underscore the
importance of broad somatic expression of *lin-35(+)* for maintaining germline function, espe-
cially under moderate temperature stress.

## Discussion

In this study, we have shown that brood size in *lin-35* mutants displays temperature sensitivity
that goes beyond what is seen in wild type. We demonstrated that although LIN-35 has a small
role in the soma to protect fertility, expression of LIN-35 in the germ line is of primary impor-
tance for fertility, especially under conditions of moderate temperature stress. Interestingly,

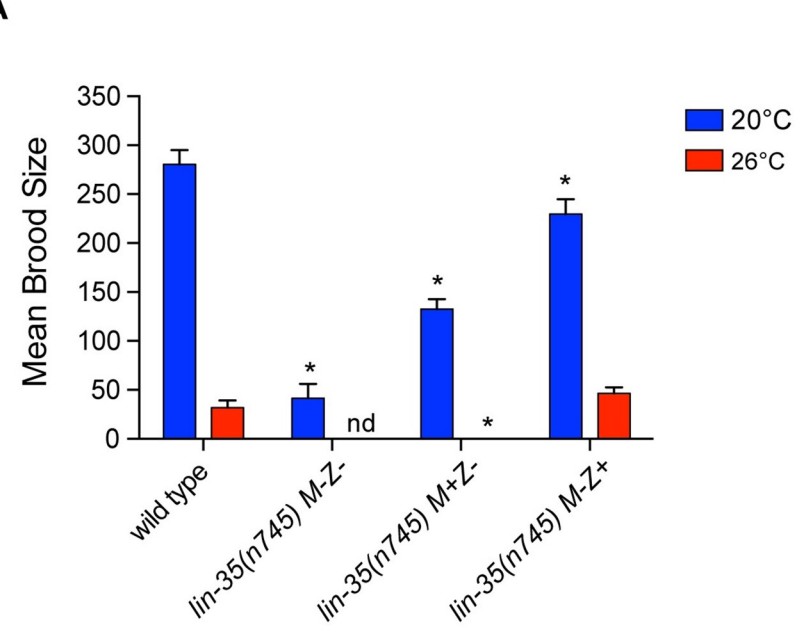

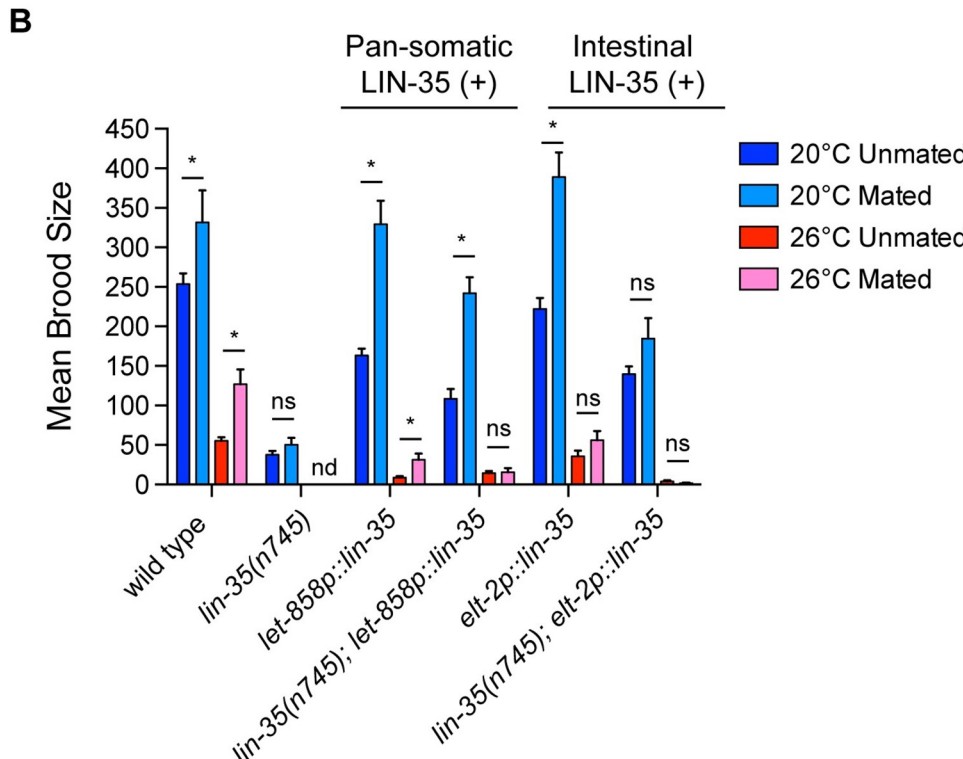

**Fig 3. Fertility defects in *lin-35* mutants are not maternal effect and are not strongly rescued by mating under moderate temperature stress.** (A) Mean brood size of fertile worms for each treatment. Presence or absence of maternally loaded *lin-35(+)*/LIN-35(+) indicated by M+/M- respectively; presence or absence of zygotically expressed *lin-35(+)* indicated by Z+/Z- respectively. Maternal load of *lin-35(+)* was not sufficient for fertility to be maintained in *lin-35* M+Z- mutants at 26°C and zygotic expression of *lin-35* in M-Z+ mutants rescued fertility loss in mutants at

26˚C. M-Z- animals did not inherit any wild-type *lin-35* products (protein or RNA) from their mother and did not have a wild-type copy of the *lin-35* gene, M+Z- animals inherited *lin-35* products from their mother but did not have a wild-type copy of the *lin-35* gene, and M-Z+ did not inherit any wild-type *lin-35* products (protein or RNA) from their mother but did have a wild-type copy of the *lin-35* gene. * significantly different than wild type at the same temperature. *P*-value ≤ 0.05 using one-way ANOVA with Turkey correction. Error bars indicate ± SEM. (B) Mean brood size of fertile worms for each genotype under the different mating and growth conditions (continual growth of hermaphrodites at 20˚C (blue) and continual growth at 26˚C (red)). Mating with wild-type males did not significantly increase the brood size of *lin-35* mutants expressing either somatic transgene at high temperature. * significantly different from same strain unmated at the same temperature. *P*-value ≤ 0.05 using students t-test. Error bars indicate ± SEM. For mated experiments, males were always grown to adulthood at 20˚C and then mating occurred at the temperature of the hermaphrodite treatment.

unlike most other LIN-35 functions, zygotic and not maternal expression of LIN-35 in the germ line was primarily important for maintaining brood size. Finally, we found that LIN-35 functions in the soma to ensure proper oocyte formation and/or function under moderate temperature stress. Our data adds to the growing understanding of the importance of LIN-35 in several stress response pathways [30], and underscores a new role for LIN-35 in regulating germline function during moderate temperature stress.

Somatic expression of LIN-35(+) partially rescued brood size in *lin-35* mutants grown from the L1 stage at either 20˚C or 26˚C. The generation of progeny relies on many somatic tissues, including intestines, somatic gonads, and neurons [14, 15, 31]. Given that *lin-35* mutants display a broad range of effects on different tissues (including slowed growth), it is not surprising that somatic rescue of LIN-35(+) could have a positive effect on brood size. LIN-35 function solely in the intestine appears to be particularly important for brood size at 20˚C, but is not sufficient to overcome fertility deficits of *lin-35* mutants at 26˚C. Marked differences in rescue between intestinally and pan-somatically expressed LIN-35(+) were seen with many phenotypes assessed, including brood size, percentage of fertile worms, sperm localization, and oocyte number. Although the somatic gonad is a necessary for fertility, expression of LIN-35(+) in both the somatic gonad and intestine was not sufficient to rescue the low brood size seen in *lin-35* mutants. The other somatic tissues in which in LIN-35(+) expression is required to maintain brood size during moderate temperature stress remain to be determined.

Pan-somatic expression of LIN-35(+) in *lin-35* mutants rescued brood size in animals upshifted to 26˚C at the L1 stage, but did not restore brood size or the percentage of the fertile population when the upshift occurred at the L4 stage. What underlies these differences? There is other evidence in support of short and more acute temperature stresses at the L4/adult stage having stronger effects on germline biology. For example, short periods of growth at increased temperatures can increase asynapsis likely due to disruptions to the physical properties of the synaptonemal complex [32, 33]. When *C. elegans* are kept at increased temperatures for longer periods they can acclimate to the conditions. Therefore, we propose that somatic expression of LIN-35(+) supports the correct development of somatic tissues during extended periods of developmental temperature stress to help maintain fertility. However, expression of LIN-35(+) in the soma is less effective at mitigating the effects of short and acute temperature stresses, which may more directly affect germline processes and therefore would require LIN-35(+) expression in the germ line to mitigate these effects.

Our somatic rescue experiments relied on multi-copy transgenic arrays, which can result in overexpression of transgenes. Therefore, we cannot rule out that overexpression of LIN-35(+) in the soma may contribute to the level of the rescue of fertility associated phenotypes observed. However, we have presented strong evidence that full fertility during moderate temperature stress is dependent on the expression of LIN-35(+) in somatic tissues outside of the intestine. First, pan-somatic expression of LIN-35(+) only weakly rescued the defects in brood

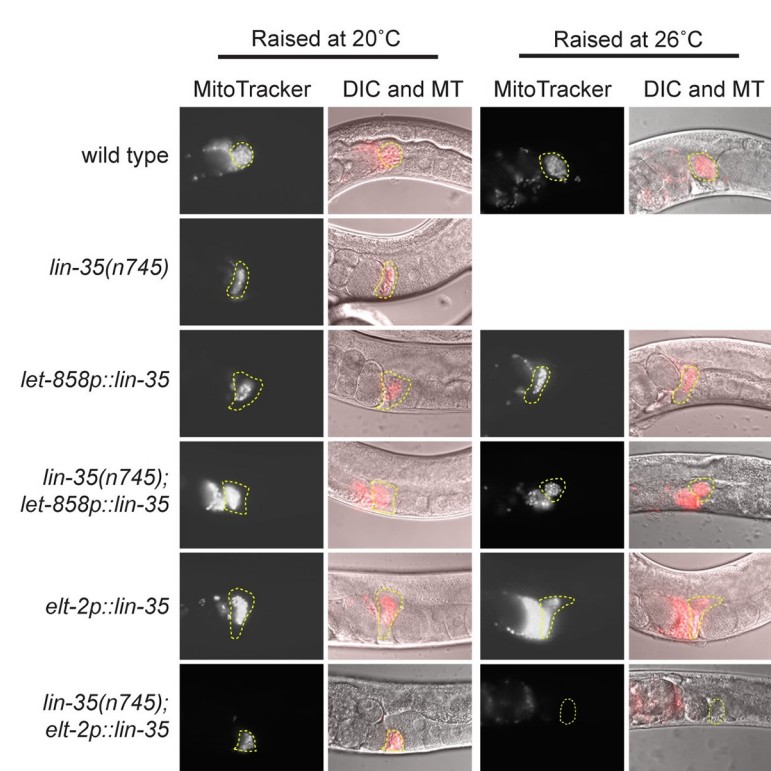

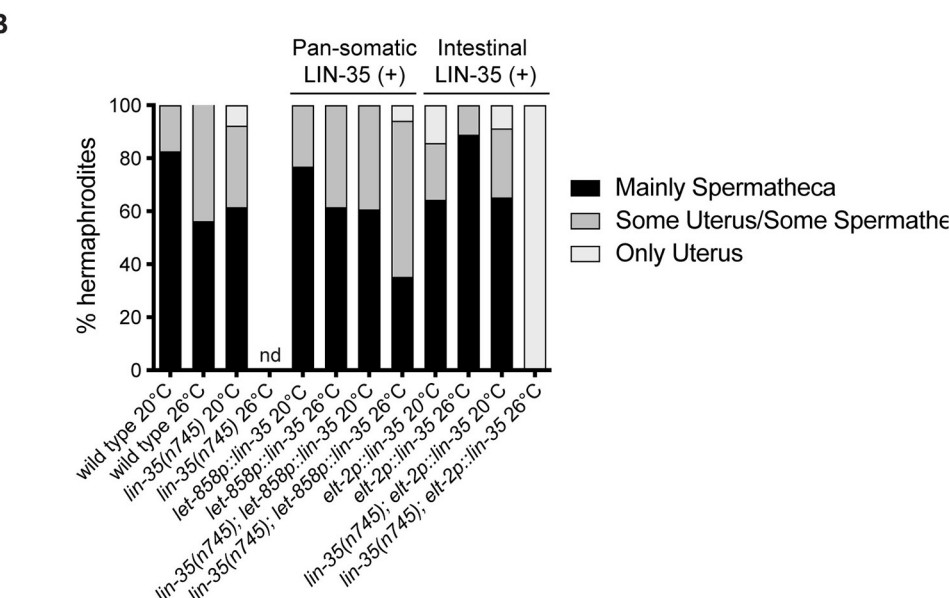

**Fig 4. Male sperm do not migrate to the spermatheca in *lin-35* mutants expressing the intestinal transgene during moderate temperature stress.** (A) Representative images of spermatheca in hermaphrodites after mating with males stained with MitoTracker Red CMXRos. The location of spermatheca is indicated by yellow outlines. (B) Percentage of hermaphrodites that show different distributions of male sperm within the reproductive tract as primarily within the spermatheca (black), a combination of sperm within the spermatheca and in the uterus (dark grey), or sperm only within the uterus (light grey). Notably no *lin-35(n745); elt-2p::lin-35* hermaphrodites have sperm in the spermatheca when grown at 26°C.

**A**

*lin-35(n745); elt-2p::lin-35*

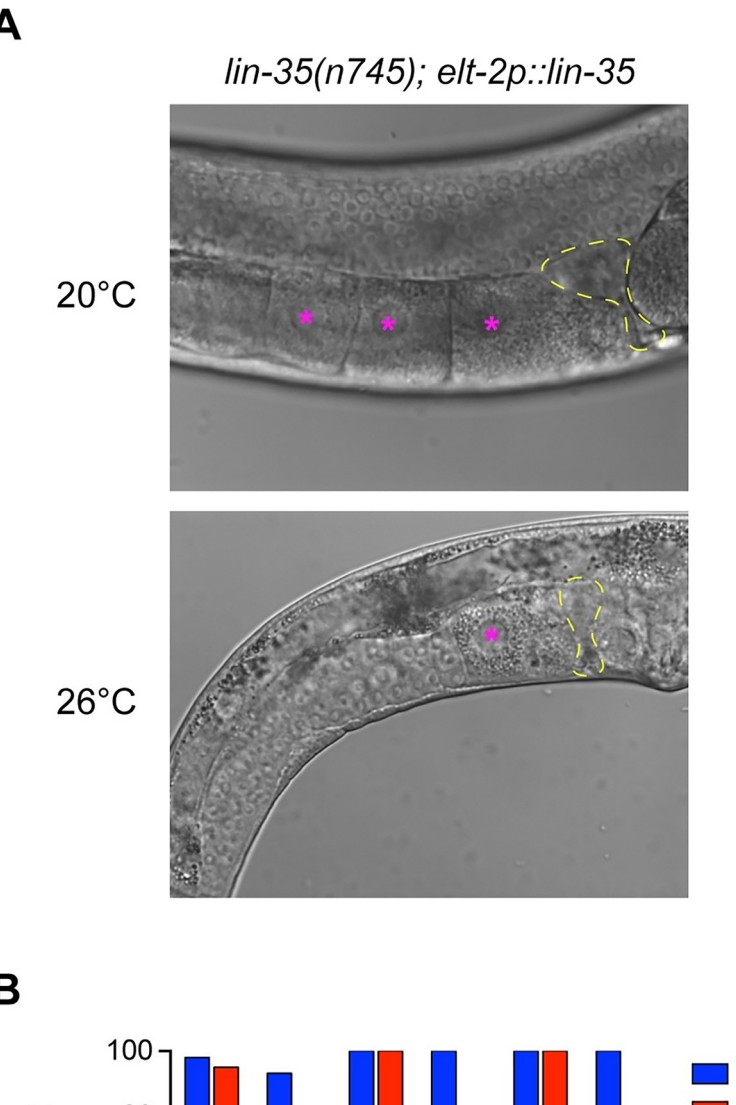

**B**

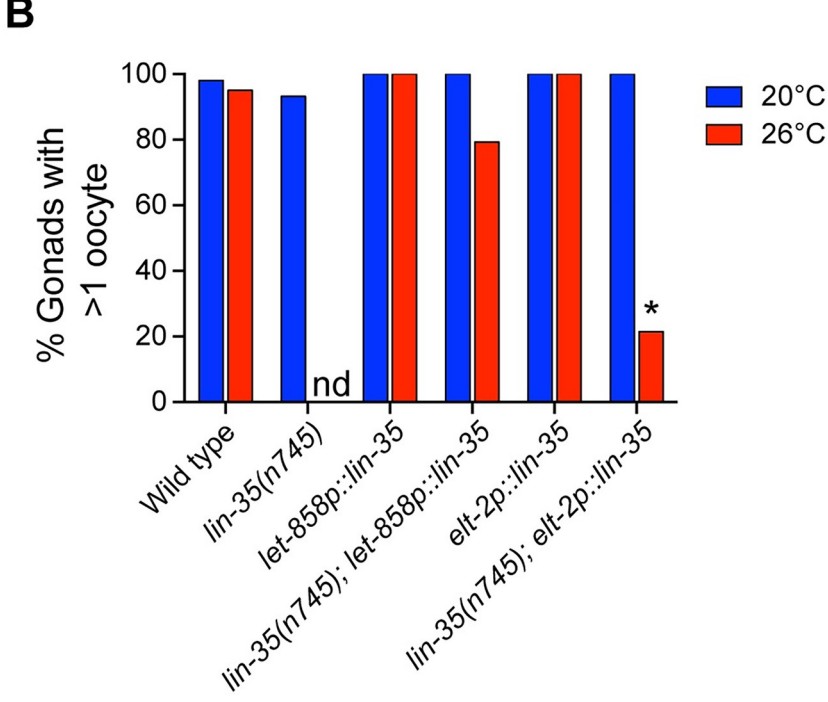

**Fig 5.** *lin-35* **mutants expressing the intestinal transgene have fewer oocytes under moderate temperature stress.**
(A) Representative DIC images of *lin-35(n745); elt-2p::lin-35* hermaphrodite gonads at 20˚C and 26˚C. The location of spermatheca is indicated by yellow outlines and oocytes are represented by a pink asterisk. (B) The percentage of gonad arms in hermaphrodites that have >1 oocyte. Significantly fewer *lin-35(n745); elt-2p::lin-35* hermaphrodites have >1 oocyte at 26˚C than at 20˚C. *significantly different from same strain at 20˚C. *P*-value ≤ 0.05 using Fisher's exact test.

size but strongly rescued other phenotypes, such as slow growth and high temperature larval arrest. This suggests that the level of somatic expression in a *lin-35* mutant background was not detrimental to somatic development. Second, the transgenes used for germline rescue of fertility were low-copy number and genomically integrated [22]. As these transgenes could strongly rescue brood size under all temperature treatments, but not provide complete rescue, it is likely that somatic expression of LIN-35(+) is important for some fertility phenotypes. Therefore, expression of LIN-35(+) in somatic tissues is likely necessary to fully restore brood size in *lin-35* mutants.

A range of germline associated phenotypes and broad disruptions in germline gene expression are seen in *lin-35* mutants [5, 12, 34], making it difficult to define specific roles for LIN-35 (+) in the germ line. The only well-defined role of LIN-35(+) the germ line is in its promotion of germline apoptosis. A slightly lower level of physiological apoptosis is seen in *lin-35* mutants under non-stress conditions [6], but these animals fail to increase the rate of germline apoptosis under various stress conditions, including DNA damage and starvation [6, 35]. Germline apoptosis is also induced under specific elevated temperature conditions [19]. Therefore, a decreased level of germline apoptosis may contribute to the reductions in fertility seen in *lin-35* mutants, especially under stress conditions. Our data also reveals a role for LIN-35(+) in the formation and/or function of mature oocytes, which appeared to be especially sensitive to temperature when LIN-35(+) was only present in the intestine. Finally, increased fertility defects are present in double mutants between *lin-35* and members of the MuvB core of the DREAM complex [34]. Thus, interactions between LIN-35 within the DREAM complex may be important for regulating gene expression in the germ line. Further research is needed to define the targets of a LIN-35/MuvB Core complex in the germ line and the impact temperature may have on the expression of these targets.

Many phenotypes associated with *lin-35* mutants are notably temperature sensitive. For example, *lin-35* mutant larvae only arrest at 26˚C, and increased temperature enhances the ectopic expression of germline expressed genes in somatic tissues and the synthetic multivulva phenotype seen in *lin-35* mutants [13, 20]. In this study, we have shown that both brood size and oocyte formation in *lin-35* mutants are also temperature sensitive, expanding the temperature sensitive nature of *lin-35* phenotypes to the germ line. Historically, temperature sensitive phenotypes are most clearly explained by a hypomorphic allele of a gene that produces a modestly changed protein, the folding of which is destabilized at increased temperatures. However, the *lin-35(n745)* allele contains a premature stop codon, has no visible protein expression, and has generally been categorized as a putative null allele [3]. Therefore, it seems unlikely that loss of functional LIN-35 protein is what ties these temperature sensitive phenotypes together. Another possible explanation could be that a protein complex in which LIN-35 normally interacts becomes unstable during moderate temperature stress when LIN-35 is absent. The most well characterized complexes involving LIN-35 are the E2F complex and the DREAM complex [34, 36, 37]. Previous work has shown that loss of other members of the E2F complex cause distinctly different germline phenotypes and lead to different changes in gene expression in mutant germ lines [12]. No studies to date have investigated the formation of the DREAM complex in the germ line. However, even at 20˚C in the soma there is a significant loss of the

DREAM complex binding to its targets in *lin-35* mutants, suggesting that phenotypes associated with loss of the complex are not temperature sensitive [34]. Increased loss of DREAM complex binding could occur at 26˚C, but this is yet to be explored. Therefore, the exact nature of the temperature sensitivity of germline phenotypes in *lin-35* mutants remains unclear.

In summary, we have shown that fertility in animals lacking LIN-35 is highly sensitive to moderate temperature stress, which is primarily a zygotic effect. LIN-35 is required in the germ line for fertility, and broadly needed within somatic tissue for oocyte formation and/or function. Thus, we provide additional evidence for the central role that LIN-35 plays in protecting animals against conditions of stress.

## Supporting information

**S1 Fig. *let-858* driven expression of LIN-35::GFP is not found in germline cells.** (A) L1 animals at 20˚C display broad expression of LIN-35::GFP in somatic tissues, but not in the the two primordial germ cells Z2/Z3 (shown by staining with anti-PGL-1). Z2/Z3 outlined with a yellow-dashed line. (B) Images of adult gonad tissue beside adult intestine show expression of LIN-35::GFP in the somatic distal tip cell (double-arrow), somatic gonad sheath cells (arrows), and intestinal cells (arrow heads) but not in either mitotic or meiotic germ cells. The gonad tissues are surrounded by yellow-dashed lines.
(TIF)

**S2 Fig. *let-858* driven expression of LIN-35::GFP represses expression of wild-type LIN-35 protein.** Western blot using anti-LIN-35 antibodies shows expression of both the wild-type LIN-35 protein and LIN-35::GFP tagged protein expressed from somatic transgenes in animals grown at either 20˚C or 26˚C. In wild-type worms expressing the *let-858p*::*lin-35*::*GFP* transgene, only LIN-35::GFP protein can be seen, while in wild-type worms containing the *elt-2p*::*lin-35*::*GFP* transgene, both LIN-35 and LIN-35::GFP protein can be seen. Anti-Tubulin was used on the same blot as a loading control. * non-specific background bands.
(TIF)

## Acknowledgments

Manuscript editors Brent Neumann and Julian Heng (Remotely Consulting, Australia) provided professional English-language editing of this article (Manuscript Certificate No. 1Ag1Uw2D).

## Author Contributions

**Conceptualization:** Brian P. Mikeworth, Lisa N. Petrella.

**Data curation:** Lisa N. Petrella.

**Formal analysis:** Brian P. Mikeworth, Frances V. Compere, Lisa N. Petrella.

**Funding acquisition:** Lisa N. Petrella.

**Investigation:** Brian P. Mikeworth, Frances V. Compere, Lisa N. Petrella.

**Methodology:** Brian P. Mikeworth, Lisa N. Petrella.

**Project administration:** Lisa N. Petrella.

**Resources:** Lisa N. Petrella.

**Supervision:** Lisa N. Petrella.

**Validation:** Brian P. Mikeworth, Frances V. Compere, Lisa N. Petrella.

**Visualization:** Brian P. Mikeworth, Frances V. Compere.

**Writing – original draft:** Lisa N. Petrella.

**Writing – review & editing:** Brian P. Mikeworth, Frances V. Compere, Lisa N. Petrella.

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
