## [Decision Letter · Decision Letter 0]

8 Jan 2023

PONE-D-22-34075LIN-35 is necessary in both the soma and germline for preserving fertility in Caenorhabditis elegans under moderate temperature stressPLOS ONE

Dear Dr. Petrella,

Thank you for submitting your manuscript to PLOS ONE. After careful consideration, we feel that it has merit but does not fully meet PLOS ONE’s publication criteria as it currently stands. Therefore, we invite you to submit a revised version of the manuscript that addresses the points raised during the review process.

We look forward to receiving your revised manuscript.

Kind regards,

Myon-Hee Lee, Ph.D

Academic Editor

PLOS ONE

Journal Requirements:

Reviewers' comments:

Reviewer's Responses to Questions

**Comments to the Author**

1. Is the manuscript technically sound, and do the data support the conclusions?

Reviewer #1: Yes

Reviewer #2: Yes

2. Has the statistical analysis been performed appropriately and rigorously? 

Reviewer #1: Yes

Reviewer #2: Yes

3. Have the authors made all data underlying the findings in their manuscript fully available?

Reviewer #1: Yes

Reviewer #2: Yes

4. Is the manuscript presented in an intelligible fashion and written in standard English?

Reviewer #1: Yes

Reviewer #2: Yes

5. Review Comments to the Author

Reviewer #1: The authors have investigated the soma and germline requirements of the C. elegans RB homolog LIN-35 in fertility under normal growth conditions and mild temperature stress.

Using transgenes expressing lin-35 under a pan-somatic promoter or an intestinal promoter, there were able to show significant rescue effects of the brood size at 20° and 26°, as well as in upshift conditions. However, no rescue of the fertility is observed at 26°. Relative fertility is fully rescued in stressed (26°) condition compared to upshift conditions using the pan-somatic transgene. This tells that some lin-35(+) somatic activity is missing when lin-35 is exclusively expressed in the intestine. The authors added lin-35(+) expression in the somatic gonad to the one in the intestine, however, it did not increase the brood size. Altogether, lin-35 somatic expression partially rescues decreased brood size and fertility associated with temperature stress, but a germline intrinsic lin-35(+) activity might also be necessary for a fully functional germline. To test this, lin-35 is expressed under the control of the pie-1 or the mex-5 promoter. Both promoters show a rescue effect and relative fertility shows a complete rescue of the stress caused by upshift of the temperature.

Then authors studied the effect of the maternal and the zygotic contributions of lin-35(+) on the brood size. Interestingly zygotic expression is sufficient to rescue the brood size to a wild type level at 26°. They further show that normal sperm is not able to compensate for all the defects, although they appear to localize properly in the spermatheca, except when lin-35(+) is expressed from an intestinal promoter at 26°. The authors also looked at oocyte formation and observed that lin-35 mutants expressing lin-35(+) under an intestinal promoter have a reduced number of oocytes at 26°. However, the pan-somatic promoter rescue almost completely the oocyte deficiency associated with lin-35 mutant. The authors conclude that LIN-35 is required both in the germline and in somatic tissue for the full reproduction capacity. Somatic LIN-35, but more predominantly germline LIN-35 are required for the germ cells to resist to mild temperature stress. The results are further interpreted in a discussion that is interesting, well written and supported by the literature on LIN-35 function that the authors appear to know thoroughly. Altogether, this study provides new insights in the function of LIN-35/RB in germline stress resistance and the maintenance of reproduction capacity.

Comments

-A large part of the results relies on rescue levels of brood size and fertility. However, levels of rescue using extra-chromosomal arrays are difficult to interpret. The MoSCI strategy from the Frøkjær-Jensen lab would have been more appropriate to avoid over-expression. Indeed, when the transgenes are expressed in wild type worms, they have a toxic effect on fertility which makes the results even more difficult to interpret.

-The authors could comment on why they are not using the lin-35 promoter instead of the let-858 promoter?

-Along the same line: why are the authors not testing the full rescue potential of a lin-35 transgene as a proof of concept?

-DIC images provided, as well as the drawings of the worms (Figure 1) are of poor quality, and I hope that there will be of better resolution in the final version.

-Figure 5 is labeled Figure 6

Reviewer #2: Previous studies have shown that C. elegans LIN-35, a pocket protein related to mammalian retinoblastoma protein (Rb), promotes fertility. LIN-35 is widely expressed throughout the body and functions in many tissues; at least some aspects of the lin-35 phenotype are temperature sensitive. The authors have a long-standing interest in germline stress sensitivity. Here, they investigate whether fertility is influenced by LIN-35 expression somatic tissues as well as the germline itself, and if this (these) function(s) is (are) particularly important under conditions of temperature stress. A complication in analyzing temperature effects in the germline is that even wildtype brood size drops at higher culture temperatures. The authors take pains to distinguish between this phenomenon and a lin-35-specific effect on brood size.

The authors evaluated germline function by determining brood size and % fertile animals, referring to the latter phenotype by the generic term, “fertility.”

The authors examine somatic involvement using transgenic strains where lin-35(+) are expressed under control of an intestinal or pan-somatic promoter in an otherwise lin-35(-) background. These transgenes rescue the lin-35(-) larval arrest phenotype, allowing strains to be maintained at high temperature. In contrast, they were evaluated germline function using germline-specific transgenes that could not rescue larval lethality, and hence they had to evaluate high temperature effects by upshifting larvae to high temperature once they were past the temperature-sensitive period.

The authors describe expression patterns of the various transgenic lines as “data not shown,” but they should present those data (in a supplement).

The authors present convincing evidence that both somatic transgenes can partially rescue the brood size defect – but not % fertile animals - at 20°C. Detailed comparisons show rescue was more complete with pan-somatic expression than with intestinal expression, indicating that lin-35 expression in multiple somatic tissues positively impacts brood size. Consistent with this, the fertility defect in animals grown at 26°C was better rescued when the pan-somatic transgene was present. They also assayed animals upshifted from 20° to 26°C as L4 larvae, and in this case somatic lin-35 expression did not rescue these animals. The authors’ interpretation of this result was not clear.

Germline expressed transgenes partially rescued the brood size defect in animals raised at 20°C and animals upshifted to 26°C. Rescue was better than with somatic transgenes, indicating germline-intrinsic expression is critical.

One confounding factor in these studies is that the somatically expressed transgenes are multicopy and present on extrachromosomal arrays, meaning they are very likely over-expressed compared to wildtype in those tissues where the promoter is active. These differences in expression level in transgenic lines may complicate some of the results, e.g., better suppression by intestinal expression in some cases and pan-somatic expression in others. The authors should include some discussion of this point.

The authors investigated whether the germline phenotype reflected a maternal effect, as is the case for some other aspects of the lin-35 phenotype. They show that maternal expression contributes to brood size at 20°C, but zygotic expression is necessary in animals upshifted to 26°C. Moreover, brood sizes are restored to wt level by a lin-35(+) allele delivered from the male.

The authors also identify sperm migration and oogenesis defects that likely contribute to the reduced brood size (sperm defect) and reduced % fertile animals (sperm and oocyte defects) at elevated temperatures. Intriguingly, the oogenesis defect can be rescued by pan-somatic lin-35 expression.

Overall, the data presented are interesting, original, and support the authors’ conclusions. This manuscript will be of interest to germline researchers and those studying the impacts of stress on development.

Specific comments

The authors’ terminology is confusing at times. Specifically, they routinely use the term “fertility” to indicate % fertile animals in a population. However, sometimes they use it to indicate brood size. For example, on page 13, the authors present data indicating paternal rescue of the brood size, saying “At 26°C lin-35(n745) M-Z+ herms have an average brood that was sig different than wildtype herms (Fig. 3A). Thus, zygotic expression alone can strongly rescue fertility of lin-35 mutants at low temp and completely rescue fertility at high temps.” The very next statement is the section heading “Mating does not significantly rescue fertility in lin-35 mutants…” I suggest the authors use a more specific term when talking about the % fertile animals in a population, for example, “Mating does not significantly rescue percent fertile adults in lin-35 mutant populations under most conditions”. There are other examples…

When introducing maternal vs embryonic expression, it would be helpful if the authors defined the “M” and “Z” terms for readers who might be new to the terminology.

Typos and awkwardly worded sentences are sprinkled throughout the manuscript.

6. PLOS authors have the option to publish the peer review history of their article (what does this mean?). If published, this will include your full peer review and any attached files.

Reviewer #1: **Yes: **Chantal Wicky

Reviewer #2: No

---

## [Author Response · Author response to Decision Letter 0]

27 Apr 2023

Please see the individual response to the editor and reviewers below.

Journal Requirements:

We made sure all the formatting meets the journals style requirements.

R00GM98436 and R15GM122005 to L.N.P these are two correct grants. They are in the Funding Source section. 

All the raw data and images have been archived through the Marquette University ePublication database and can be found at https://epublications.marquette.edu/mikeworth_2023/

This information was provided with submission and in the Materials and Methods section of the revised document.

These raw gels have been added to the public archive of the raw data in the section called Figure S2.

We have created a new supplemental figure (now Figure S1) that shows the images of the expression of the transgene in question in soma and not in germline tissue. 

All supporting data captions have been moved to the end of the manuscript and are properly cited in the results section.

We have reviewed our references per the instructions

Reviewers' comments:

Reviewer's Responses to Questions

Comments to the Author

1. Is the manuscript technically sound, and do the data support the conclusions?

Reviewer #1: Yes

Reviewer #2: Yes

2. Has the statistical analysis been performed appropriately and rigorously?

Reviewer #1: Yes

Reviewer #2: Yes

3. Have the authors made all data underlying the findings in their manuscript fully available?

Reviewer #1: Yes

Reviewer #2: Yes

4. Is the manuscript presented in an intelligible fashion and written in standard English?

Reviewer #1: Yes

Reviewer #2: Yes

5. Review Comments to the Author

Reviewer #1: The authors have investigated the soma and germline requirements of the C. elegans RB homolog LIN-35 in fertility under normal growth conditions and mild temperature stress.

Using transgenes expressing lin-35 under a pan-somatic promoter or an intestinal promoter, there were able to show significant rescue effects of the brood size at 20° and 26°, as well as in upshift conditions. However, no rescue of the fertility is observed at 26°. Relative fertility is fully rescued in stressed (26°) condition compared to upshift conditions using the pan-somatic transgene. This tells that some lin-35(+) somatic activity is missing when lin-35 is exclusively expressed in the intestine. The authors added lin-35(+) expression in the somatic gonad to the one in the intestine, however, it did not increase the brood size. Altogether, lin-35 somatic expression partially rescues decreased brood size and fertility associated with temperature stress, but a germline intrinsic lin-35(+) activity might also be necessary for a fully functional germline. To test this, lin-35 is expressed under the control of the pie-1 or the mex-5 promoter. Both promoters show a rescue effect and relative fertility shows a complete rescue of the stress caused by upshift of the temperature.

Then authors studied the effect of the maternal and the zygotic contributions of lin-35(+) on the brood size. Interestingly zygotic expression is sufficient to rescue the brood size to a wild type level at 26°. They further show that normal sperm is not able to compensate for all the defects, although they appear to localize properly in the spermatheca, except when lin-35(+) is expressed from an intestinal promoter at 26°. The authors also looked at oocyte formation and observed that lin-35 mutants expressing lin-35(+) under an intestinal promoter have a reduced number of oocytes at 26°. However, the pan-somatic promoter rescue almost completely the oocyte deficiency associated with lin-35 mutant. The authors conclude that LIN-35 is required both in the germline and in somatic tissue for the full reproduction capacity. Somatic LIN-35, but more predominantly germline LIN-35 are required for the germ cells to resist to mild temperature stress. The results are further interpreted in a discussion that is interesting, well written and supported by the literature on LIN-35 function that the authors appear to know thoroughly. Altogether, this study provides new insights in the function of LIN-35/RB in germline stress resistance and the maintenance of reproduction capacity.

Comments

-A large part of the results relies on rescue levels of brood size and fertility. However, levels of rescue using extra-chromosomal arrays are difficult to interpret. The MoSCI strategy from the Frøkjær-Jensen lab would have been more appropriate to avoid over-expression. Indeed, when the transgenes are expressed in wild type worms, they have a toxic effect on fertility which makes the results even more difficult to interpret.

We agree with the reviewer that in hindsight this would have been a better method. But we do not have the resources to redo all the experiments with newly made lines. However, we still feel like the overall outcomes of the paper stand even given the caveats raised by the reviewer. We have changed some of the writing throughout to better express the caveats with using extrachromosomal arrays- specifically see the new writing in the Discussion on lines 527-541.

-The authors could comment on why they are not using the lin-35 promoter instead of the let-858 promoter?

We did not use the endogenous lin-35 promoter because it drives expression in all tissues including the germline normally. While it may have worked to use the endogenous promoter in an array that was silenced in the germline we decided to use let-858 instead which had a known history of driving broad somatic expression and not germline expression when used on arrays. 

-Along the same line: why are the authors not testing the full rescue potential of a lin-35 transgene as a proof of concept?

Previous work from another lab has already published data on the use of the endogenous lin-35 and showed it rescued the fertility and larval arrest phenotypes (Kudron et al. 2013). We were specifically interested instead on what happens when lin-35 was only present in somatic or germline tissues. We’ve added language to clarify this which can be found on line 213-215 in the revised manuscript. 

-DIC images provided, as well as the drawings of the worms (Figure 1) are of poor quality, and I hope that there will be of better resolution in the final version.

On our end all the figures are crisp and meet all figure quality requirements. We aren’t sure how to change this and our inquiries to the editor did not result in a clear answer. If issues with resolution persist, we would work with the editor on them.

-Figure 5 is labeled Figure 6

This has been fixed

Reviewer #2: Previous studies have shown that C. elegans LIN-35, a pocket protein related to mammalian retinoblastoma protein (Rb), promotes fertility. LIN-35 is widely expressed throughout the body and functions in many tissues; at least some aspects of the lin-35 phenotype are temperature sensitive. The authors have a long-standing interest in germline stress sensitivity. Here, they investigate whether fertility is influenced by LIN-35 expression somatic tissues as well as the germline itself, and if this (these) function(s) is (are) particularly important under conditions of temperature stress. A complication in analyzing temperature effects in the germline is that even wildtype brood size drops at higher culture temperatures. The authors take pains to distinguish between this phenomenon and a lin-35-specific effect on brood size.

The authors evaluated germline function by determining brood size and % fertile animals, referring to the latter phenotype by the generic term, “fertility.”

The authors examine somatic involvement using transgenic strains where lin-35(+) are expressed under control of an intestinal or pan-somatic promoter in an otherwise lin-35(-) background. These transgenes rescue the lin-35(-) larval arrest phenotype, allowing strains to be maintained at high temperature. In contrast, they were evaluated germline function using germline-specific transgenes that could not rescue larval lethality, and hence they had to evaluate high temperature effects by upshifting larvae to high temperature once they were past the temperature-sensitive period.

The authors describe expression patterns of the various transgenic lines as “data not shown,” but they should present those data (in a supplement).

See new Figure S1 for these images.

The authors present convincing evidence that both somatic transgenes can partially rescue the brood size defect – but not % fertile animals - at 20°C. Detailed comparisons show rescue was more complete with pan-somatic expression than with intestinal expression, indicating that lin-35 expression in multiple somatic tissues positively impacts brood size. Consistent with this, the fertility defect in animals grown at 26°C was better rescued when the pan-somatic transgene was present. They also assayed animals upshifted from 20° to 26°C as L4 larvae, and in this case somatic lin-35 expression did not rescue these animals. The authors’ interpretation of this result was not clear.

We thank the reviewer for pointing out this oversight. We have added a paragraph to the discussion to address these differences in somatic rescue between the two types of temperature stress, see lines 513-526.

Germline expressed transgenes partially rescued the brood size defect in animals raised at 20°C and animals upshifted to 26°C. Rescue was better than with somatic transgenes, indicating germline-intrinsic expression is critical.

One confounding factor in these studies is that the somatically expressed transgenes are multicopy and present on extrachromosomal arrays, meaning they are very likely over-expressed compared to wildtype in those tissues where the promoter is active. These differences in expression level in transgenic lines may complicate some of the results, e.g., better suppression by intestinal expression in some cases and pan-somatic expression in others. The authors should include some discussion of this point.

We have added a paragraph to the discussion to address the potential issues surrounding complications with the transgenes, see lines 527-541.

The authors investigated whether the germline phenotype reflected a maternal effect, as is the case for some other aspects of the lin-35 phenotype. They show that maternal expression contributes to brood size at 20°C, but zygotic expression is necessary in animals upshifted to 26°C. Moreover, brood sizes are restored to wt level by a lin-35(+) allele delivered from the male.

The authors also identify sperm migration and oogenesis defects that likely contribute to the reduced brood size (sperm defect) and reduced % fertile animals (sperm and oocyte defects) at elevated temperatures. Intriguingly, the oogenesis defect can be rescued by pan-somatic lin-35 expression.

Overall, the data presented are interesting, original, and support the authors’ conclusions. This manuscript will be of interest to germline researchers and those studying the impacts of stress on development.

Specific comments

The authors’ terminology is confusing at times. Specifically, they routinely use the term “fertility” to indicate % fertile animals in a population. However, sometimes they use it to indicate brood size. For example, on page 13, the authors present data indicating paternal rescue of the brood size, saying “At 26°C lin-35(n745) M-Z+ herms have an average brood that was sig different than wildtype herms (Fig. 3A). Thus, zygotic expression alone can strongly rescue fertility of lin-35 mutants at low temp and completely rescue fertility at high temps.” The very next statement is the section heading “Mating does not significantly rescue fertility in lin-35 mutants…” I suggest the authors use a more specific term when talking about the % fertile animals in a population, for example, “Mating does not significantly rescue percent fertile adults in lin-35 mutant populations under most conditions”. There are other examples…

We have revised the manuscript to specify “brood size” or “percentage of fertile animals” to point to specific experiments, while removing the general term “fertility” in most contexts. We decided to us the term “fertility phenotypes” when describing the combination of the two measures scored and the general trends associated with LIN-35(+) expression.

When introducing maternal vs embryonic expression, it would be helpful if the authors defined the “M” and “Z” terms for readers who might be new to the terminology.

We added additional language to make this clearer in the text, see the paragraph wat lines 363-377.

Typos and awkwardly worded sentences are sprinkled throughout the manuscript.

We edited for typos and awkward sentences and contracted with a copy editor to further refine the editing (see Acknowledgment section for specific editors used).

6. PLOS authors have the option to publish the peer review history of their article (what does this mean?). If published, this will include your full peer review and any attached files.

Do you want your identity to be public for this peer review? For information about this choice, including consent withdrawal, please see our Privacy Policy.

Reviewer #1: Yes: Chantal Wicky

Reviewer #2: No

---

## [Decision Letter · Decision Letter 1]

26 May 2023

LIN-35 is necessary in both the soma and germline for preserving fertility in Caenorhabditis elegans under moderate temperature stress

PONE-D-22-34075R1

Dear Dr. Petrella,

We’re pleased to inform you that your manuscript has been judged scientifically suitable for publication and will be formally accepted for publication once it meets all outstanding technical requirements.

Kind regards,

Myon-Hee Lee, Ph.D

Academic Editor

PLOS ONE

Additional Editor Comments (optional):

Reviewers' comments:

Reviewer's Responses to Questions

**Comments to the Author**

1. If the authors have adequately addressed your comments raised in a previous round of review and you feel that this manuscript is now acceptable for publication, you may indicate that here to bypass the “Comments to the Author” section, enter your conflict of interest statement in the “Confidential to Editor” section, and submit your "Accept" recommendation.

Reviewer #1: All comments have been addressed

Reviewer #2: All comments have been addressed

2. Is the manuscript technically sound, and do the data support the conclusions?

Reviewer #1: Yes

Reviewer #2: (No Response)

3. Has the statistical analysis been performed appropriately and rigorously? 

Reviewer #1: Yes

Reviewer #2: Yes

4. Have the authors made all data underlying the findings in their manuscript fully available?

Reviewer #1: Yes

Reviewer #2: Yes

5. Is the manuscript presented in an intelligible fashion and written in standard English?

Reviewer #1: Yes

Reviewer #2: Yes

6. Review Comments to the Author

Reviewer #1: Dear Authors,

Thanks for addressing my comments. The data is interpreted in an adequate way and the manuscript reads well now.

Reviewer #2: The revised manuscript, PONE-D-22-34075R1, from Mikeworth et al. addresses all my comments on the initial submission. The authors now discuss the implications of -and caveats to - many findings that were previously unaddressed. The writing throughout is much clearer and more straight forward than on the original submission.

7. PLOS authors have the option to publish the peer review history of their article (what does this mean?). If published, this will include your full peer review and any attached files.

Reviewer #1: **Yes: **Chantal Wicky

Reviewer #2: No

---

## [Editor Report · Acceptance letter]

2 Jun 2023

PONE-D-22-34075R1 

LIN-35 is necessary in both the soma and germline for preserving fertility in *Caenorhabditis elegans* under moderate temperature stress 

Dear Dr. Petrella:

I'm pleased to inform you that your manuscript has been deemed suitable for publication in PLOS ONE. Congratulations! Your manuscript is now with our production department. 

Kind regards, 

on behalf of

Dr. Myon-Hee Lee 

Academic Editor

PLOS ONE